# Being Both a Parent and a Healthcare Worker in the Pandemic: Who Could Be Exhausted More?

**DOI:** 10.3390/healthcare9050564

**Published:** 2021-05-11

**Authors:** Güzin Çakmak, Zeynel Abidin Öztürk

**Affiliations:** Department of Internal Medicine, Division of Geriatric Medicine, Faculty of Medicine, Gaziantep University, TR-27310 Gaziantep, Turkey; zabidin@gantep.edu.tr

**Keywords:** anxiety, COVID-19, health care professionals, parenting stress index

## Abstract

(1) Background: The COVID-19 pandemic has changed the living conditions of many people. Many people felt significantly constrained. However, for individuals who are both parents and healthcare professionals, the situation seems more troubling in other ways. (2) Objectives: Based on this, we planned a study evaluating demographic characteristics, COVID-19-related anxiety levels, and parenting-related stress levels of the health care professionals who were working in the University Hospital. We also evaluated the parameters that affect COVID-19-related anxiety and parenting stress. (3) Methods: The level of COVID-19-related anxiety is assessed by the coronavirus anxiety scale. The parenting stress index-short form is used for evaluation of parenting stress. Statistical analysis was done by SPSS version 22. (4) Results: Female gender, working as a nurse, a history of COVID-19, and having a child attending daycare were parameters that increased the level of COVID-19-related anxiety. Occupation, being a parent of a schoolchild and/or primary school child, being a parent of a child receiving face-to-face education, and having more than two children were found to be risk factors for parenting stress. Clinically significant parenting stress was found to increase threefold in healthcare workers with more than two children (R^2^ = 0.101, *p* = 0.039). (5) Conclusions: Healthcare professionals, who are also parents, play a huge role both at home and in the hospital in the pandemic. Therefore, it is inevitable that their stress and anxiety levels increase. It is important to determine the factors that cause stress and anxiety and to take measures in this direction to get through this process well.

## 1. Introduction

Coronavirus disease 2019 (COVID-19) is an acute respiratory infection that can progress with pneumonia and acute respiratory distress syndrome (ARDS) and caused by Severe Acute Respiratory Syndrome Coronavirus-2 (SARS-CoV-2) [1]. The first case was originated in Wuhan, China, at the end of 2019. In March 2019, the disease had spread around the world and was accepted as a pandemic [2]. As of 8 January 2021, the number of COVID-19 cases worldwide was 88,203,229, the number of deaths was 1,901,510, and the number of people recovered was 49,203,004 [3]. It is difficult to control the disease because of the high contagiousness and the fact that asymptomatic carriers are not noticed. Healthcare workers are among the risk groups such as the elderly, those with chronic diseases, cancer patients, and immunosuppressive patients. Moreover, healthcare workers are at very significant risk of being an asymptomatic carrier and they may contribute spread of the disease [4]. It is estimated that around 10–20% of all diagnosed as COVID-19 are healthcare workers [5]. Healthcare workers can cause nosocomial transmission or spread the disease at home. Since the beginning of the pandemic, children and youth also have been considered as a potential cause of disease spread. The reason for this is the higher rate of asymptomatic disease in young people. As a result, schools were closed, and distance education was started in many countries [6]. In this case, being the child of healthcare professionals becomes a privileged concept. These children and young people can act as a bridge between two risky areas in terms of infection spread, such as hospitals and schools. Healthcare workers who are concerned about many issues such as being sick, spreading the disease, and not being able to provide good care to patients also become worried about infecting their children if they are parents. All these suggest that healthcare workers with children may have an increase in anxiety levels and parenting stress levels during this period. Psychological distress that parents experience while they are trying to do the parental duty is defined as parenting stress [7]. In this study, we aimed to reveal the factors affecting the coronavirus-related anxiety levels and parenting stress index levels of healthcare professionals who work during the pandemic period.

## 2. Methods

### 2.1. Participants and Study Design

One-hundred-and-sixty-two professional health workers in Gaziantep University Hospital were included in this cross-sectional study. These participants were asked to fill out printed questionnaires. Data were recorded by a staff member who was not being informed about the study design. First, socio-demographic data such as age, gender, employment, age of first parenthood, number of children, and school status of the children were questioned. All participants were asked if they were suffered from COVID-19. The anxiety levels of the participants about COVID-19 were evaluated with the coronavirus anxiety scale (Appendix A) [8]. Parenting stress was assessed by the parenting stress index-short form [7].

### 2.2. Exclusion Criteria

Those with a past or present psychiatric diagnosis were excluded from the study.

### 2.3. Coronavirus Anxiety Scale

The Coronavirus Anxiety Scale (CAS) is a short questionnaire prepared to recognize possible causes of dysfunctional anxiety associated with the COVID-19 pandemic by Lee et al. [8]. This scale is a Likert type scale consisting of 5 questions. Each question is scored between 0 and 4 by the participant. The lowest score that can be obtained from the survey is 0 and the highest score is 20. The Turkish validity and reliability study of this scale was conducted by Bicer et al. [9]. It was observed that the correlation coefficients for each question ranged between 0.625 and 0.784. In our study, the Cronbach alpha correlation coefficient for all questions is 0.895.

### 2.4. Parenting Stress Index

The level of parenting stress was assessed with the Parenting Stress Index-Short Form (PSI-SF) scale, which was developed by Abidin et al. [7]. This scale is a Likert type scale consisting of 36 questions. Each question is scored between 1 and 5 by the participant. The lowest score that can be obtained from the survey is 36 and the highest score is 180. The test includes 3 sub-scales. These are the Parental Distress subscale (PD-SF), the Parent-Child Dysfunctional Interaction subscale (PCDI-SF), and The Difficult Child subscale (DC-SF). Each sub-scale consists of 12 questions. Values between 81 and 89 percentile for total PSI-SF score, PD-SF score and DC-score indicate the high-stress presence, and values between 90 and 100 percentile indicate the presence of clinically significant stress. Values between 81 and 84 percentile for total PCDI-SF score indicate the high-stress presence, and values between 85 and 100 percentile indicate the presence of clinically significant stress [10]. The Turkish adaptation of the index was done by Mert et al. [10]. Cronbach alpha correlation coefficient of total PSI-SF score was 0.71. The correlation coefficient of PD-SF score was 0.81, PCDI-SF 0.76 and DC-SF 0.78. In our study, the correlation coefficient of total stress scores was 0.947. The correlation coefficient of scores of three subscales were 0.886, 0.916, and 0.920, respectively.

### 2.5. Statistical Analysis

The variables were analyzed for their distribution normality using the Kolmogorov–Smirnov and Shapiro–Wilk test. All data were disturbed normally (*p* > 0.05). Power analysis was done by Gpower 3.9.1 software. To find statistically significant, the expectation that a medium effect size (dz = 0.5) will occur between the parameters, the minimum number required was determined as 70 (α = 0.05; 1 − β = 0.80). Descriptive statistics are given for continuous variables. Continuous variables of groups were assessed by using the independent sample t-test and one-way analysis of variance (ANOVA). The data were expressed as mean ± deviation (S.D.). We used multinominal logistic regression to simulate a model to determine factors affecting clinically significant parenting stress. The statistical significance level was determined as *p* < 0.05. We used SPSS version 22.0 (IBM, Armonk, NY, USA) to analyze the data.

## 3. Results

One-hundred and sixty-two health care workers who were parents and employed at Gaziantep University Hospital were evaluated in this study. Ninety-two of them (56.8%) were male, and 70 of them (43.2%) were female. Their mean age was 36.9 ± 6.8. Twenty-six of them were physician, 61 of them were nurses, and 75 of them were cleaning staff. Thirty-seven of the participants (22.8%) had suffered from COVID-19. The socio-demographic characteristics, CAS scores, and PSI-SF scores of both genders are summarized in Table 1. CAS scores were higher in women than in men (*p* = 0.004). Health care workers who had suffered from COVID-19 had higher CAS scores than others (*p* = 0.033). CAS scores were higher in nurses than in cleaning staff (*p* = 0.019). The CAS scores of healthcare workers with children attending daycare were also higher than others (*p* = 0.032). The parameters affecting the CAS score are summarized in Table 2. According to the total PSI-SF score, 131 people were normal, 14 were highly stressed, and 17 people were clinically highly stressed. According to the PD-SF score, 129 people were normal, 16 people were highly stressed, and 17 people were clinically highly stressed. According to the DC-SF score, 129 people were normal, 15 people were highly stressed, and 18 people were clinically highly stressed. According to the PCDI-SF score, 129 people were normal, 7 people were highly stressed, 26 people were clinically highly stressed.

The PSI-SF, PCDI-SF score, and DC-SF score were higher in those with school-age children (*p* = 0.028, *p* = 0.039, *p* = 0.029). The total PSI-SF score was higher in healthcare workers with children attending primary school (*p* = 0.049). The PCDI-SF score was higher in cleaning staff than physicians and nurses (*p* = 0.036, *p* = 0.03). The PCDI-SF score and DC-SF score were higher in health care workers who had more than 2 children (*p* = 0.006, *p* = 0.003). The total PSI-SF, PCDI-SF, and DC-SF scores were higher in health care workers with children that receiving face-to-face education (*p* = 0.022, *p* = 0.039, *p* = 0.041). We found that number of children was independently related to clinically significant parenting stress in multinominal logistic regression analysis (R^2^ = 0.101, *p* = 0.039). We found that clinically significant parenting stress was three times more common in healthcare workers with more than two children. The parameters affecting the total PSI-SF are summarized in Table 3. Results of multinominal logistic regression analysis were summarized in Table 4. The parameters affecting the PSI-SF subscale scores are summarized in Table 5.

## 4. Discussion

It is known that pandemics can negatively affect the psychological well-being of large populations [11]. Healthcare workers are at serious risk for pandemic-related psychological problems. In a recent study, the prevalence of traumatic stress was 73.4%, depression 50.7%, anxiety 44.7%, and insomnia 36.1% among Chinese healthcare workers who faced with the COVID-19 pandemic [12]. In this study, we evaluated the anxiety associated with the pandemic with CAS. It was previously stated that CAS scores were associated with psychosomatic symptoms, extreme hopelessness, alcohol, or drug use, and suicidal tendency [9]. Therefore, it is important to know the parameters that affect the CAS score to prevent negative situations that may be encountered. In our study, it was observed that the female gender had more COVID-19 related anxiety. In a study by Özdin et al., female gender revealed as a risk factor for pandemic-related anxiety in the general population. They evaluated anxiety by Hospital Anxiety and Depression Scale (HADS) and the Health Anxiety Inventory (HAI) [13]. CAS scores were also found to be high in those who suffered from COVID-19. Mazza and colleagues evaluated COVID-19 survivors for psychological symptoms one month after hospital discharge. Post-traumatic stress disorder was observed in 28%, depression 31%, anxiety 42%, obsessive-compulsive symptoms 28%, and insomnia 40% in these individuals. In a meta-analysis, the prevalence of post-traumatic stress disorder was revealed to be 32.2%, depression was 14.9%, and anxiety disorders was 14.8% [14]. It was not surprising that COVID-19 survivors particularly experienced post-traumatic stress disorder and anxiety symptoms. There is no previous study on this subject regarding healthcare professionals. However, the increased anxiety in healthcare workers after COVID-19 can be interpreted as they are not expecting to develop immunity. The higher anxiety level associated with coronavirus in nurses compared to cleaning staff could be attributed to their closer contact with patients. Szepietowski et al. have revealed that there is no difference between the anxiety and depression status of healthcare professionals working in the dermatology department and nephrology department during the pandemic process. In this study, there was also no difference in mental health status between nurses and doctors [15]. Unlike this, Vizheh et al. reported that nurses, female employees, front line healthcare workers, young health workers, and those working at high risk of infection show more psychological symptoms than other healthcare workers in their systematic review. However, they also concluded that secondary traumatic stress was observed more in nurses that were not in front line and general population than front line nurses [16]. Parents of children attending daycare were revealed to be more anxious about COVID-19. This can be explained by the fact that younger children are more dependent on parents and serious difficulty in providing care occurs because of the closure of daycare centers.

Parental stress could be affected by “job, environment, marital relationships, and daily problems” of parents as well as children’s characteristics. In our study, parental stress was found to be higher in parents with school-age children, primary school children, and those whose children received face-to-face education. This can be attributed to the unstable situation of schools for being open or closed. When schools are open, their cause of anxiety is the risk of infection. When schools are closed, it is particularly difficult for working parents to monitor children’s distance education process and ensure their home safety during working hours. Hiraoka et al. reported that parental stress increased after schools were closed [17]. Since the parents in our study were healthcare workers, their concerns about contaminating their children may be higher. On the other hand, we carried out this study in a period when the entire curriculum was given by distance education, but face-to-face education was given for a limited time depending on the approval of the family. Thus, we had the chance to compare the parents of children who received face-to-face training with those who could not. Disruption in the parent-child relationship and “difficult child” problems were observed higher in parents with more than two children. It is possible to associate this situation with a decrease in the time that parents will spare for each child. As the number of children increases, financial resources will have to be shared among children. Moreover, healthcare workers with more than two children had higher total parenting stress scores and clinically significant parenting stress was more prevalent. There is no other study in the literature showing that the number of children has such an effect on parenting stress. This situation can be attributed to the fact that the population in which the study was conducted was healthcare workers with serious time limitations. The parent–child relationship is seen to be more problematic in cleaning staff than physicians and nurses. It could be possible to associate this situation with low-income levels. A previous study showed that financial problems in the parents of pediatric cancer patients may be associated with parental stress [18]. Coronavirus-related anxiety was also seemed to increase parental distress, in this study. Brown et al. also showed that anxiety and depression are associated with parenting stress during the pandemic period [19].

In the pandemic, being a parent and being a healthcare worker are situations that could be stressful. We cannot ignore that it is difficult to do these two difficult tasks at the same time. People develop various methods to cope with the stress caused by the COVID-19 pandemic. These methods may be for solving the problem or for relieving stress. Activities to solve the problem are used more frequently by healthcare professionals. Man et al. observed that healthcare professionals use planning refocusing and positive re-evaluation methods more than the general population to cope with stress [20]. Providing institutional support as well as personal efforts will be useful in this process. Institutions should also support their employees through nutritional support, organizing working hours, solving accommodation and transportation problems, providing adequate personal protective equipment, training on stress management, providing psychological support, and providing medical support to sick healthcare workers [21].

Hyun et al. emphasized the importance of psychosocial support in pandemic management [22]. Identifying situations that both increase coronavirus-induced stress and increase parenting stress could be helpful when planning support for those who fulfil this difficult task.

Our study is important because there has been no previous study evaluating coronavirus-related stress and parenting stress together in parents who are healthcare professionals. The main limitation of the study was the size of the population studied. If our study could be done on a larger population, we could have achieved more results. It could also be good to know the PSI-SF scores of the participants before the pandemic. We tried to overcome this problem by excluding those with a psychiatric diagnosis.

## 5. Conclusions

During the pandemic period, gender, being suffered from COVID-19, “number, age and education of children”, and income levels are seemed to affect the difficulty level of being a parent. It would be helpful to pay more attention to these risks when planning government subsidies and institutional supports. The increased anxiety among women employees, nurses, those who have suffered from COVID-19 and those who have children going to daycare also suggests that increasing psychosocial support for these groups may be beneficial. It could be useful to do more studies on this subject.

## Figures and Tables

**Table 1 healthcare-09-00564-t001:** Socio-demographic characteristics of health care workers.

Characteristics	Female (*n* = 70) (43.2%)	Male (*n* = 92) (56.8%)	*p*
Age (mean ± SD)	37.3 ± 6.5	36.6 ± 6.1	0.503
Number of children (mean ± SD)	1.9 ± 0.68	2.1 ± 0.9	0.164
CAS score (mean ± SD)	2.8 ± 4.4	1.2 ± 2.4	0.004 *
PSI-SF score (mean ± SD)	70.9 ± 26.5	73.3 ± 25.4	0.548
PD-SF (mean ± SD)	27 ± 11	27.5 ± 10.1	0.767
PCDI Score (mean ± SD)	20.3 ± 9.4	21.8 ± 10.6	0.371
DC-SF (mean ± SD)	23.2 ± 9.8	2.1 ± 10.5	0.573
Age of first parenthood (mean ± SD)	26.8 ± 3.7	27.5 ± 3.9	0.25
Presence of chronic disease (*n*) (%)	25 (35.7%)	20 (21.7%)	0.021 *
History of COVID-19 (*n*) (%)	18 (25.7%)	19 (20.7%)	0.112
Occupation			
Cleaning staff (75)	16 (22.9%)	59 (64.1%)	<0.001 *
Nurse (61)	47 (67.1%)	14 (15.2%)	
Physician (25)	7 (10%)	19 (20.7%)	

PD-SF: Parental Distress Subscale-Short Form. PCDI-SF: Parent–Child Dysfunctional Interaction Subscale-Short Form. DC-SF: Difficult Child Subscale-Short Form. COVID-19: Coronavirus disease 2019. *: Statistically significant.

**Table 2 healthcare-09-00564-t002:** Parameters affecting CAS scores.

Parameters	CAS Score (mean ± SD)	*p*
Gender (*n*)		0.004 *
Female (70)	2.8 ± 4.4	
Male (92)	1.2 ± 2.4	
Work (n)		0.048 *
Cleaning staff (75)	1.4 ± 2.6	
Nurse (61)	2.8 ± 4.7	
Physician (25)	1.5 ± 1.4	
Children attending daycare (n)		0.032 *
Yes	3.7 ± 5.4	
No	1.7 ± 3.2	
COVID-19 history		0.033 *
Yes (37)	3 ± 4.9	
No (125)	1.6 ± 2.9	

CAS: Coronavirus anxiety scale. COVID-19: Coronavirus disease 2019. *: Statistically significant.

**Table 3 healthcare-09-00564-t003:** Parameters affecting total PSI-SF score.

Parameters	PSI-SF (mean ± SD)	*p*
Occupation (n)		0.130
Cleaning staff (75)	76.4 ± 28.1	
Nurse (61)	67.5 ± 24.9	
Physician (25)	71.6 ± 18.8	
School-age children		0.028 *
Yes (111)	75.3 ± 27.3	
No (51)	65.7 ± 21.2	
Primary school children		0.049 *
Yes (100)	75.4 ± 28.1	
No (62)	67.2 ± 21.1	
High school-university school children		0.999
Yes (42)	72.27 ± 21.8	
No (120)	72.27 ± 27	
Face to face education		0.022 *
Yes (77)	77.2 ± 27.8	
No (85)	67.9 ± 23.2	
Distance education		0.798
Yes (36)	73.3 ± 25.2	
No (126)	72 ± 26.1	
Having more than two children		0.014 *
Yes (41)	80.8 ± 31.5	
No (121)	69.4 ± 23	

*: Statistically significant.

**Table 4 healthcare-09-00564-t004:** Parameters affecting presence of clinically significant parenting stress.

Variables	B	Wald	Exp(B)	*p*
Face-to-face education	0.131	0.045	1.140	0.832
Number of children	1.143	4250	3.135	0.039 *
Primary school children	0.745	0.990	2.106	0.320

*: Statistically significant.

**Table 5 healthcare-09-00564-t005:** Parameters affecting PSI-SF subscale scores.

Parameters	Parental Distress (mean ± SD)	*p*	Parent–Child Dysfunctional Interaction (mean ± SD)	*p*	Difficult Child (mean ± SD)	*p*
Work (n)						
Cleaning staff (75)	28.4 ± 10.7	0.314	23.3 ± 11.4	0.034*	24.7 ± 11.2	0.341
Nurse (61)	25.7 ± 10.7		19.6 ± 9.3		22.2 ± 9.4	
Physician (25)	27.8 ± 9.3		18.4 ± 6.3		24.2 ± 8.4	
School-age children						
Yes (111)	28 ± 10.8	0.241	22.25 ± 10.7	0.039*	24.9 ± 10.6	0.029*
No (51)	25.9 ± 9.6		18.7 ± 8.4		21.1 ± 8.6	
Primary school children						
Yes (100)	28.4 ± 11.1	0.087	22.04 ± 10.9	0.144	24.6 ± 10.9	0.113
No (62)	25.5 ± 9.2		19.6 ± 8.5		22 ± 8.6	
High school-university school children						
Yes (42)	25.49 ± 8.5	0.229	22 ± 7.9	0.578	24.8 ± 8.9	0.426
No (120)	27.9 ± 10.9		20.9 ± 10.7		23.3 ± 10.5	
Face to face education						
Yes (77)	28.6 ± 11	0.149	22.9 ± 10.9	0.039*	25.4 ± 11	0.041*
No (85)	26.2 ± 9.9		19.6 ± 9.2		22.13 ± 9.1	
Distance education						
Yes (36)	27.44 ± 10	0.93	21.8 ± 10.1	0.651	24 ± 9.3	0.826
No (126)	27.3 ± 10.6		20.9 ± 10.1		23.6 ± 10.4	
Having more than two children						
Yes (41)	28.2 ± 10.8	0.510	24.9 ± 12.5	0.006*	27.6 ± 12.1	0.003*
No (121)	26.9 ± 10.4		19.9 ± 8.9		22.3 ± 9	

PSI-SF: Parenting Stress Index-Short Form. *: Statistically significant.

## Data Availability

The data presented in this study are available on request from the corresponding author.

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
