# Peer review of "Being Both a Parent and a Healthcare Worker in the Pandemic: Who Could Be Exhausted More?"

_healthcare, 2021, doi:10.3390/healthcare9050564_

Round 1

Reviewer 1 Report

The study adresses an important issue on the health of health workers in this hospital during the COVID pandemic.

Was any approval of an ethic commitee granted or no necessary?

It would have been intersting to compare the health workers to a general population in Poland (now in the literature).

And athough the study seems properly conducted, he would have beeen interesting to make more references or comparision with general population and in other comparable countries. It would be intersting to see the questionnaire. The conclusion seems a little bit short

Author Response

Healthcare,

Submission date: 18 January 2021

Manuscript title:
Being both a parent and a healthcare worker in the pandemic: Who could be exhausted more?

Manuscript ID: healthcare-1097522

Reply to reviewer 1,

Dear reviewer,

I am grateful for the time and effort you put into evaluating the work. We made some additions to our article in line with your suggestions. I will try to explain them to you in this post.

Comment-1: Was any approval of an ethic commitee granted or no necessary?

Answer to comment-1:

We thought that the approval of the ethics committee was not required, as no name and surname information was included in the questionnaires and the questionnaires were not answered face to face. I guarantee that no medical intervention has been made to any participant regarding the study, only the questionnaires distributed to the participants by a neutral person were filled out by the participants.

Comment-2: It would have been intersting to compare the health workers to a general population in Poland (now in the literature).

Answer to comment-2:

We have added a reference to the discussion regarding this suggestion.

“Szepietowski et al. have revealed that there is no difference between the anxiety and depression status of healthcare professionals working in the dermatology department and nephrology department during the pandemic process. In this study, there was also no difference in mental health status between nurses and doctors.”

Reference: Szepietowski JC, Krajewski P, BiÅ‚ynicki-Birula R, PoznaÅ„ski P, Krajewska M, Rymaszewska J, et al. Mental health status of health care workers during the COVID-19 outbreak in Poland: One region, two different settings. Dermatol Ther 2020;33:1–4. https://doi.org/10.1111/dth.13855. 

Comment-3:And although the study seems properly conducted, he would have beeen interesting to make more references or comparision with general population and in other comparable countries. It would be intersting to see the questionnaire. The conclusion seems a little bit short.

Answer-3: We have added a reference to the discussion regarding this suggestion.

“Unlike this, Vizheh et al. reported that nurses, female employees, front line healthcare workers, young health workers, and those working at high risk of infection show more psychological symptoms than other healthcare workers in their systematic review. However, they also concluded that secondary traumatic stress was observed more in nurses that were not in front line and general population than front line nurses.”

Reference: Vizheh M, Qorbani M, Arzaghi SM, Muhidin S, Javanmard Z, Esmaeili M. The mental health of healthcare workers in the COVID-19 pandemic: A systematic review. J Diabetes Metab Disord 2020;19:1967–78. https://doi.org/10.1007/s40200-020-00643-9

We added the Coronavirus anxiety scale as appendix. I had provided the PSI-SF questionnaire as a validated Turkish questionnaire. The English form is available for a fee and is not included in similar articles, so I could not get it, but I can buy it if it is mandatory.

We expanded the conclusion part upon your suggestion.

Conclusion:

During the pandemic period, gender, being suffered from COVID-19, “number, age and education of children”, and income levels are seemed to affect the difficulty level of being a parent. It would be helpful to pay more attention to these risks when planning government subsidies and institutional supports. The increased anxiety among women employees, nurses, those who have suffered from COVID-19 and those who have children going to daycare also suggests that increasing psychosocial support for these groups may be beneficial. It could be useful to do more studies on this subject.

Questionnaire-1: Coronavirus anxiety scale

How often have you experienced the following activities over the last 2 weeks?

Not at all (0)        

Rare, less than a day or two (1)

Several days (2)

More than 7 days (3)

Nearly every day over the last 2 weeks (4)

  1. I felt dizzy, lightheaded, or faint, when I read or listened to news about the coronavirus.

0             1             2              3             4

  1. I had trouble falling or staying asleep because I was thinking about the coronavirus.

0             1             2              3             4

  1. I felt paralyzed or frozen when I thought about or was exposed to information about the coronavirus.

0             1             2              3             4

  1. I lost interest in eating when I thought about or was exposed to information about the coronavirus.

0             1             2              3             4

  1. I felt nauseous or had stomach problems when I thought about or was exposed to information about the coronavirus.

0             1             2              3             4

We have incorporated changes that reflect the detailed suggestions you have graciously provided. We also hope that our edits and the responses we provide below satisfactorily address all the issues and concerns you have noted.

Sincerely,

Guzin Cakmak MD

Gaziantep University Faculty of Medicine, Department of Internal Medicine, Division of Geriatric Medicine

drguzincakmak@gmail.com

Tel: +905457621951

Reviewer 2 Report

Although studies about the effect of SARS-CoV-2 pandemics in health workers are frequent, this study, in particular, gave us novelty when presenting parent-related stress.

As a suggestion, the discussion can be enriched with some considerations about the implementation of specific mental health programs for health workers.

Author Response

Healthcare,

Submission date: 18 January 2021

Manuscript title:
Being both a parent and a healthcare worker in the pandemic: Who could be exhausted more?

Manuscript ID: healthcare-1097522

Reply to reviewer 2,

Dear reviewer,

I am grateful for the time and effort you put into evaluating the work. We made some additions to our article in line with your suggestions. I will try to explain them to you in this post.

Comment-1: As a suggestion, the discussion can be enriched with some considerations about the implementation of specific mental health programs for health workers.

Answer to comment-1:

I have added references to the discussion regarding this suggestion.

“People develop various methods to cope with the stress caused by the COVID-19 pandemic. These methods may be for solving the problem or for relieving stress. Activities to solve the problem are used more frequently by healthcare professionals. Man and colleagues observed that healthcare professionals use planning refocusing and positive re-evaluation methods more than the general population to cope with stress.”

Reference: Man MA, Toma C, Motoc NS, Necrelescu OL, Bondor CI, Chis AF, et al. Disease perception and coping with emotional distress during covid-19 pandemic: A survey among medical staff. Int J Environ Res Public Health 2020;17:1–13. https://doi.org/10.3390/ijerph17134899

“Providing institutional support as well as personal efforts will be useful in this process. Institutions should also support their employees through nutritional support, organizing working hours, solving accommodation and transportation problems, providing adequate personal protective equipment, training on stress management, providing psychological support, and providing medical support to sick healthcare workers.”

Reference: Walton M, Murray E, Christian MD. Mental health care for medical staff and affiliated healthcare workers during the COVID-19 pandemic. Eur Hear J Acute Cardiovasc Care 2020;9:241–7. https://doi.org/10.1177/2048872620922795.

“Hyun et al. emphasized the importance of psychosocial support in pandemic management”

Reference: Hyun J, You S, Sohn S, Kim SJ, Bae J, Baik M, et al. Psychosocial support during the COVID-19 outbreak in Korea: Activities of multidisciplinary mental health professionals. J Korean Med Sci 2020;35:1–13. https://doi.org/10.3346/JKMS.2020.35.E211.

We have incorporated changes that reflect the detailed suggestions you have graciously provided. We also hope that our edits and the responses we provide below satisfactorily address all the issues and concerns you have noted.

Sincerely,

Guzin Cakmak MD

Gaziantep University Faculty of Medicine, Department of Internal Medicine, Division of Geriatric Medicine

drguzincakmak@gmail.com

Tel: +905457621951
